# Evaluation of the Efficacy of CPP-ACP Remineralizing Mousse in Molar-Incisor Hypomineralized Teeth Using Polarized Raman and Scanning Electron Microscopy—An *In Vitro* Study

**DOI:** 10.3390/biomedicines10123086

**Published:** 2022-12-01

**Authors:** Inês Cardoso-Martins, Sofia Pessanha, Ana Coelho, Sofia Arantes-Oliveira, Paula F. Marques

**Affiliations:** 1Faculdade de Medicina Dentária da Universidade de Lisboa, Rua Professora Teresa Ambrósio, 1600-277 Lisbon, Portugal; 2NOVA School of Science and Technology, Campus Caparica, 2829-516 Caparica, Portugal

**Keywords:** Raman spectroscopy, electron scanning microscopy, molar–incisor hypomineralization, tooth remineralization, dental enamel, enamel, pediatric dentistry

## Abstract

Remineralization of tooth enamel can be achieved by applying a complex of casein phosphopeptides and amorphous calcium phosphate (CPP-ACP). However, the efficacy and optimization of this agent in molar–incisor hypomineralization (MIH) lacks evidence. The purpose of this study is to evaluate the efficacy of CPP-ACP tooth mousse in remineralizing MIH-affected enamel in an optimized 28-day protocol using polarized Raman microscopy and scanning electron microscopy. The protocol was applied to two types of MIH opacities, white and yellow, and compared against sound enamel specimens before and after treatment. Data was analyzed using a one-way ANOVA and LSD post hoc multiple comparisons test (*p* < 0.05) for the Raman analysis. Hypomineralized enamel showed an improvement of its structure after CPP-ACP supplementation. In addition, Raman spectroscopy results showed a decrease in the depolarization ratio of the symmetric stretching band of phosphate (*p* < 0.05 for both groups). In conclusion, there was an improvement in mineral density and organization of the hypomineralized enamel after treatment with CPP-ACP tooth mousse.

## 1. Introduction

Molar–incisor hypomineralization (MIH) is described as the hypomineralization of one to four permanent first molars frequently in combination with affected incisors [1]. This hypomineralization occurs when systemic factors affect the ameloblastic function during the maturation phase of amelogenesis [2]. The global and European prevalence of MIH ranges from 13.1% to 14.2% [3] and 3.6 to 25% [1,4], respectively. Enamel from the MIH-affected surface is characterized by a lower mineral concentration, lower organization of the crystalline structure, and higher porosity, making those teeth more vulnerable to abrasion and carious lesions [5]. The presence of chronic pulpal inflammation in MIH teeth may be due to their increased enamel porosity and possible bacterial penetration through the large dentinal tubules, making them extremely sensitive, arduous to hygiene, and difficult to anesthetize [1,6]. In more severe cases, the porous enamel may even fracture after eruption or under masticatory forces, creating post-eruptive enamel breakdowns (PEB) [1].

Clinically, MIH teeth present clearly demarcated opacities that vary in size, shape, and color, from white to yellow/brownish [1,2]. Several studies showed that these defects have increased carbon and carbonate concentrations and higher protein content when compared to normal enamel [2,5,7,8,9]. To enhance the mechanical properties of MIH teeth, remineralization is a good treatment option that can increase the mineral content of enamel and prevent its post-eruptive enamel breakdowns [10]. The literature shows that fluoride varnish can be effective in remineralizing MIH teeth [2]. Furthermore, in a recent study, a novel remineralizing agent, a biomimetic hydroxyapatite, was successfully used in the improvement of the MIH condition [11]. The remineralization of tooth enamel can also be achieved by applying a complex of casein phosphopeptides and amorphous calcium phosphate (CPP-ACP), that acts as a reservoir of calcium and phosphate, stabilizing high concentrations of these ions [12,13,14]. There is evidence that the application of CPP-ACP cream decreases the loss of minerals, and the depth and width of lesions [14]. Several studies demonstrated that the consistent use of CPP-ACP promotes an increase in mineral density and inhibits demineralization [15,16]. In 2011, Baroni and Marchionni observed an improvement in mineralization, morphology, and porosities of MIH-affected enamel following a continued use of a cream containing 10% CPP-ACP [17]. However, there is still a limited number of studies in the literature regarding the remineralizing efficacy of this agent in MIH-affected teeth [7,13].

Raman spectroscopy has been used to evaluate the symmetric stretching mode of the tetrahedral phosphate group (PO43-, with a Raman shift of ~959 cm^−1^) and infer upon the mineral phase of enamel [18]. Fraser et al. evaluated changes in the intensity and peak shape of this Raman band, together with the ratios to the amide I and carbonate, to study teeth affected with MIH [5]. Raman spectroscopy was also used in combination with SEM to evaluate MIH-affected permanent molars through the phosphate-to-carbonate ratio [19]. Recent advances in Raman microscopy also showed the suitability, sensitivity, and specificity of polarized Raman spectroscopy for the evaluation of demineralized enamel [20]. Studies have also showed an increase in the depolarization ratio of this band in demineralized enamel, prompting the use of this methodology for carious lesion diagnosis [21,22]. Franco et al. used Raman microscopy to evaluate the effectiveness of CPP-ACP tooth mousse on the remineralization of MIH opacities after a single application [23].

According to the recent literature, there is no study that has used polarized Raman microscopy combined with SEM to evaluate the efficacy of remineralization protocols in different MIH opacities.

The aim of the present study is to understand, through polarized Raman spectroscopy analysis and SEM observation, if a 28-day protocol of application of CPP-ACP is effective in remineralizing hypomineralized enamel in two types of MIH lesions (white and yellow). The tested hypothesis was that the application of CPP-ACP did not increase the organization of the enamel apatite crystals of MIH molars in vitro.

## 2. Materials and Methods

The sample size consisted of 20 severely hypomineralized maxillary and mandibular molar teeth extracted due the presence of severe painful symptoms conditioning the patient’s quality of life. The selection of severe hypomineralized teeth was based on the EAPD diagnostic criteria of MIH [2]. An orthodontic consultation was performed, and the extractions were adapted to the patient’s orthodontic treatment plan. Upon extraction and verbal consent of the guardian of the patient, each tooth was stored and preserved in accordance with ISO conditions (ISO/TS 11405) in a refrigerated 1% chloramine solution for 1 week, and then in Hank’s solution (BioWhittaker™, Lonza, MD, USA). A control group was created using 12 sound molar teeth extracted mainly for orthodontic reasons. The control teeth were sound teeth that presented no enamel defects, cracks, caries, or fractures.

### 2.1. Sample Description and Preparation

All teeth went through a prophylactic protocol using a polishing brush with pumice paste in a low rotation handpiece (0.04 mg of pumice; 0.8 mL of water) for 60 s, before being washed with distilled water for 60 s and submitted to an ultrasound bath in ethanol 100% for 60 s using the Bransonic^®^ M2800-E, Emerson, Danbury, CT, USA, electronic cleaning device. Each tooth was encased in an acrylic block using a hot glue pistol and sectioned parallel to the occlusal surface to split the root from the crown using a water-cooled diamond-impregnated circular saw (Isomet, Buehler Ltd., Lake Bluff, IL, USA).

The SEM analysis requires metallization that makes a before–after approach unfeasible; however, Raman analyses are non-invasive and do not require specific sample preparation, so a before–after approach was undertaken. Twenty hypomineralized teeth were then sectioned perpendicular to the occlusal surface to obtain the following: one enamel opacity in sixteen teeth and two enamel opacities in four teeth. The 12 sound teeth were also sectioned perpendicular to the occlusal surface.

#### 2.1.1. Raman Spectroscopy

The 12 opacities from the MIH teeth and the 6 surfaces of the sound teeth were allocated to one of the three groups: Group A—6 white hypomineralized opacities; Group B—6 yellow hypomineralized opacities; Group C—6 sound teeth. Once prepared and whilst awaiting testing, the samples were stored fully hydrated in Hank’s solution BioWhittaker ™ (Lonza, MD, USA).

#### 2.1.2. Scanning Electron Microscopy (SEM)

The 12 opacities from the MIH teeth and the 6 surfaces of the sound teeth were allocated to one of the following 6 groups: Group D—3 white hypomineralized opacities; Group E—3 yellow hypomineralized opacities; Group F—3 sound teeth; Group G—3 white hypomineralized opacities for CPP-ACP treatment protocol; Group H—3 yellow hypomineralized opacities for CPP-ACP treatment protocol; Group I—3 sound teeth for CPP-ACP treatment protocol. Once prepared and whilst awaiting testing, the samples were stored fully hydrated in Hank’s solution (BioWhittaker™, Lonza, MD, USA). Figure 1 represents a flowchart of the study design.

### 2.2. Analysis and Treatment Protocol

#### 2.2.1. Raman Spectroscopy Analysis

Each sample from groups A, B and C was removed from the Hanks’ solution (BioWhittaker™, Lonza, MD, USA), washed with distilled water for 5 s, dried with absorbent paper, placed on a holder, and analyzed with Raman microscopy to determine the depolarization ratio before and after the CPP-ACP treatment protocol.

Polarized Raman spectra of samples were obtained using an XploRA Confocal Microscope (Horiba, Palaiseau, France) with a 785 nm laser. Using an entrance slit of 200 μm, and a confocal hole of 300 μm, the scattered light collected by the objective was dispersed onto the air-cooled CCD array of an Andor iDus detector (Oxford instruments, Bristol, UK) with a 1200 lines/mm grating. This way, the spectral range investigated was from 200 cm^−1^ to 1300 cm^−1^ with a spectral resolution of 4 cm^−1^. A 100× objective (N.A. = 0.9) was used to focus on the surface of enamel, as well as a 50% neutral density filter rendering an incident power on the sample of 4.8 ± 0.4 mW (Lasercheck^®^, Edmund Optics, Mainz, Germany). Each spectrum was obtained by 3 accumulations of 20 s each and an average of 10 spot analyses were performed for each sample. In order to determine the depolarization ratio (ρ959) of the symmetric stretching band of phosphate ions (ν1~959 cm^−1^), in each spot, spectra were recorded with cross and parallel polarization to the polarization of the incident laser. The depolarization ratio was then determined as follows:$$\rho_{959}=\frac{I_{959\;}⊥}{I_{959}∥}$$
where *I*959 II is the intensity of the Raman band at ~959 cm^−1^ using parallel polarization, and *I*959⊥ is the intensity of the Raman band at ~959 cm^−1^ using perpendicular polarization.

#### 2.2.2. SEM Analysis

Regarding the treatment groups (G, H, and I), the samples were removed from the Hank’s solution and the treatment protocol with CPP-ACP was performed daily for a period of 28 consecutive days. The samples of the control groups (D, E, and F) were kept in a balanced Hanks solution for 28 days. On day 29, all specimens from all groups were prepared according to the following protocol for debris removal and enamel prism revealing (Figure 2):
(1)Conditioning with 2 mL of 5.25% sodium hypochlorite for 60 s;(2)Washing with distilled water for 10 s;(3)Ultrasound cleaning with 100% ethanol for 60 s;(4)Washing with distilled water for 5 s;(5)Drying with air syringe for 10 s;(6)Conditioning with 2 mL of 10% phosphoric acid for 20 s;(7)Washing with distilled water for 10 s;(8)Ultrasound cleaning with 100% ethanol for 60 s;(9)Washing with distilled water for 5 s;(10)Drying with air syringe for 10 s.

After debris removal protocol application, specimens were attached to metallic supports with double sided carbon tape (NemTape, Nisshin, Japan) and sputter-coated with 200 nm gold/palladium in an argon atmosphere (Jeol, Tokyo, Japan). Specimens’ surfaces were examined using a Hitachi, SU3800 (Tokyo, Japan) scanning electron microscope. Microphotographs of representative areas at 1000×, 2000×, 5000×, and 10,000× magnifications were obtained for each specimen using Esprit software (v1.8.2.2167, Bruker, MA, EUA).

#### 2.2.3. CPP-ACP Treatment Protocol

The treatment procedure was adapted from the protocol used by Cardoso-Martins et al. and Shetty et al. for in vitro studies with CPP-ACP [24,25]. The binding of ACP to CPP is pH-dependent and decreases as the pH decreases [26]. Enamel remineralization by CPP-ACP can occur over a range of pH values from 4.5 to 7 [27]. In this protocol, the decrease in the pH by a demineralizing agent enables this remineralization process. Through the local release of calcium and phosphate ions, this nanoaggregate allows the maintenance of a supersaturated mineral state that suppresses demineralization and promotes mineralization. The treatment procedure was carried out daily for a period of 28 days [24,25]. Each of the enamel samples were treated with the remineralizing agent—10% CPP-ACP (GC Tooth mousse™, GC, Leuven, Belgium)—for a period of 2 min, following which the samples were individually immersed in 2 mL of a demineralizing solution (2.0 mMol/L calcium, 2.0 mMol/L phosphate, 75 mMol/L acetic acid, pH 4.4) for a period of 3 h. Afterwards, samples were re-treated with the remineralizing agent for 2 min. To finalize, the enamel samples were individually immersed in 2 mL of Hank’s balanced salt solution (BioWhittaker™, Lonza, MD, USA) for a period of 21 h.

The demineralizing agent was replaced every 5 days, and the Hank’s solution every 48 h.

### 2.3. Statistical Analysis

The sample size was calculated according to *n* = 1 + 2C (S/D)^2^ [28], which resulted in *n* = 6. The sample size calculation was based on results of a pilot study and used the following assumptions: the mean of differences (D = 0.0117), standard deviation (S = 0.0057), and considering the degree of significance, a = 0.05 and 90% power.

Statistical analysis was performed with SPSS v26.0, (IBM, New York, NY, USA). A Shapiro–Wilk test for normal distribution evaluation was performed. Levene’s test showed that the values obtained have equal variances. Since data distribution was normal, a one-way ANOVA and an LSD post hoc multiple comparisons test were performed. A significance level of 0.05 was considered.

## 3. Results

The Raman band of interest corresponding to the symmetric stretching of phosphate (ν1 PO43- ~959 cm^−1^) was identified. The spectra recorded for a sample presenting a hypomineralized white opacity before (Figure 3) and after (Figure 4) treatment with GC Tooth Mousse containing CPP-ACP is shown, before baseline correction.

There was an increase in the main band in the parallel mode concomitant with a decrease in the same band in the perpendicular mode, after treatment. The results for the mean depolarization ratio of the symmetric stretching band of phosphate for the three studied groups (A, B, C) are presented in Table 1.

For group A, the hypomineralized teeth with white opacities, the mean values obtained for the depolarization ratio before and after treatment were 0.029 ± 0.004 and 0.021 ± 0.003, respectively. There was a statistically significant decrease (*p* < 0.05) in the mean values of the depolarization ratio, which suggests an improvement in the enamel mineralization after treatment.

Furthermore, in group B, comprising the teeth presenting yellow hypomineralized opacities, there was a statistically significant decrease (*p* < 0.05) in the mean values of the depolarization ratio after treatment, with values ranging from 0.044 ± 0.004 before to 0.037 ± 0.008 after treatment. Therefore, there was also an improvement in the enamel mineralization with treatment in this group.

The mean values for the depolarization ratio for the sound teeth (C group) before treatment was 0.019 ± 0.002, and after treatment it was 0.017 ± 0.003. There was no statistically significant difference (*p* > 0.05) in the enamel mineralization of the sound teeth group before and after treatment. The sound teeth enamel surface presents apatite crystals methodically organized perpendicular to the surface, revealing a high anisotropy of the symmetric stretching band of phosphate. Candido et al. and Ko et al., using Raman spectroscopy, showed that the arrangement of apatite crystals is altered in demineralized enamel, since they present a more disordered structure. Accordingly, in demineralized enamel, the depolarization ratio value is higher [18,20].

The SEM images of the white and yellow opacities (D and E groups) revealed deficient or hollow rod cores. The crystals building up the enamel rods were disorganized, rock-shaped, without a uniform orientation, and difficult to distinguish. In the white opacities group, the borders of the enamel rods were dissolved with prominent inter-rod spaces, and a structureless layer was partially covering the enamel prisms (Figure 5a). In the yellow opacities group, the borders of the enamel rods were indistinct, with fewer visible inter-rod areas (Figure 6a). White and yellow opacities submitted to CPP-ACP treatment (G and H groups) showed filled rod cores with tighter packed and more well-organized crystals when compared with hypomineralized teeth not submitted to the remineralization CPP-ACP protocol, where wider inter-rod spaces were present (Figure 5b and Figure 6b)

A normal prismatic surface without structural alterations was present for both sound teeth groups. There were well-defined enamel rods with distinct borders and narrow inter-rod areas. The crystals in the enamel rods were densely packed, well organized, and presented a uniform orientation (F and I groups) (Figure 7).

## 4. Discussion

The Raman spectroscopy analysis of this study demonstrated that the treatment protocol used for CPP-ACP application had a positive and a statistically significant effect in reducing the depolarization ratio on enamel surfaces with either white or yellow hypomineralized opacities.

The SEM study showed that hypomineralized enamel of white and yellow opacities after CPP-ACP treatment protocol had an improvement on the enamel structure, that exhibited filled rod cores with tighter packed and more well-organized crystals when compared with hypomineralized teeth not submitted to remineralization protocol. The SEM images of the sound teeth enamel opacities non-submitted and submitted to CPP-ACP treatment (F and I groups) showed a normal prismatic surface.

Results from this study demonstrated that MIH opacities’ enamel structure in the white and yellow opacities (D and E groups) is characterized by having hydroxyapatite crystals more loosely and irregularly packed than in sound teeth (Figure 5a and Figure 6a). This is consistent with Raman results with a higher value of depolarization ratio for groups A and B when compared with Group C. The literature also shows that hypomineralized areas have a disordered, structureless configuration of hydroxyapatite crystals compared with sound enamel [29]. Lygidakis et al., showed that the unorganized microstructure of MIH teeth led to a reduction in mechanical properties, such as hardness and modules of elasticity [2].

White and yellow opacities (G and H groups) showed an improvement in the enamel structure after CPP-ACP treatment. Rods evolved a more mature and geometric structure, showing filled cores with tighter packed and more well-organized crystals (Figure 5b and Figure 6b). This is corroborated by Raman spectroscopy results comparing the before and after in groups A and B, which showed a decrease in depolarization ratio values and, therefore, an increase in the organization of the enamel apatite crystals after CPP-ACP treatment. Cevc et al. observed that there was a larger proportion of well-oriented microcrystals in caries-resistant teeth, indicating that the degree of hydroxyapatite microcrystal alignment is a property that may help determine their mechanical resistance [30]. The results obtained in this study are suggestive of an increased mineralization of MIH teeth after CPP-ACP treatment that could reflect an increased mechanical resistance. Baroni et al. also observed a marked improvement in enamel morphology after the application of CPP-ACP on MIH molars [17]. Franco et al. also showed an improvement in the degree of crystallite orientation in both white and yellow enamel opacities after treatment with CPP-ACP tooth mousse [23].

According to the Raman data obtained, there was a greater improvement in the organization of the enamel apatite crystals of MIH teeth after CPP-ACP treatment in the white opacities group compared to the yellow group. This difference might be explained due to the higher porosity and larger prismatic cracks of the yellow lesions when compared to the white ones [5].

The protocol with the CPP-ACP treatment was adapted from the protocol used by Cardoso-Martins et al. in an in vitro study with the same remineralizing agent assessed over 28 days in MIH teeth [24]. As a result, it was possible to decrease the samples depolarization ratio, which represents an improvement in the degree of crystallite orientation. The white opacities’ depolarization ratio mean values reached the corresponding level of the healthy teeth mean values before treatment.

A potential limitation of this study rests on the fact that it was based on an in vitro environment. Moreover, a single remineralizing agent was used and, therefore, a control group with another agent was lacking. Furthermore, a SEM–EDS (energy-dispersive X-ray spectroscopy) could have been used, but in this study Raman spectroscopy was preferred instead. There is extensive evidence that Raman spectroscopy is a very sensitive tool for the evaluation of the demineralization of enamel and that it can be used to study slight changes in the enamel structure, so we used this methodology for the evaluation of the opposite process [23]. Furthermore, due to the metallization process, a direct comparison of the Ca and P content and the depolarization ratio of phosphate would not be possible, as the sample groups were different. The small sample size must also be considered, which was due to the challenges in obtaining the specimens, since they relate to the extraction of hypomineralized permanent molars in children.

The results of the present study support the rejection of the tested hypothesis, warranting in vivo studies in MIH teeth with this remineralizing agent where administration protocols can be adjusted. In future studies, this protocol should evolve, either by increasing the number of applications or extending the duration, in order to improve the yellow opacities’ depolarization ratio, mean values and to achieve values close to the healthy teeth.

This study highlights the importance of more research using this type of remineralization protocols to improve MIH teeth properties, as well as the need for further clinical trials to support the recommendation for the use of CPP-ACP to increase the mineral content of hypomineralized teeth. In the future, it would be of interest to study new remineralizing agents, such as a biomimetic nano-hydroxyapatite based on the integration of calcium and phosphate, at the level of demineralized dental surfaces of MIH teeth [11]. Hence, managing this condition appropriately could prevent its associated negative impacts on the patient’s quality of life [2,7].

## 5. Conclusions

In this in vitro study, as determined through Raman spectroscopy analysis and SEM evaluation, there were significant improvements in the organization of MIH enamel apatite crystals after CPP-ACP treatment. In conclusion, there was a significant remineralization of enamel after a consecutive daily application of CPP-ACP tooth mousse over 28 days.

## Figures and Tables

**Figure 1 biomedicines-10-03086-f001:**
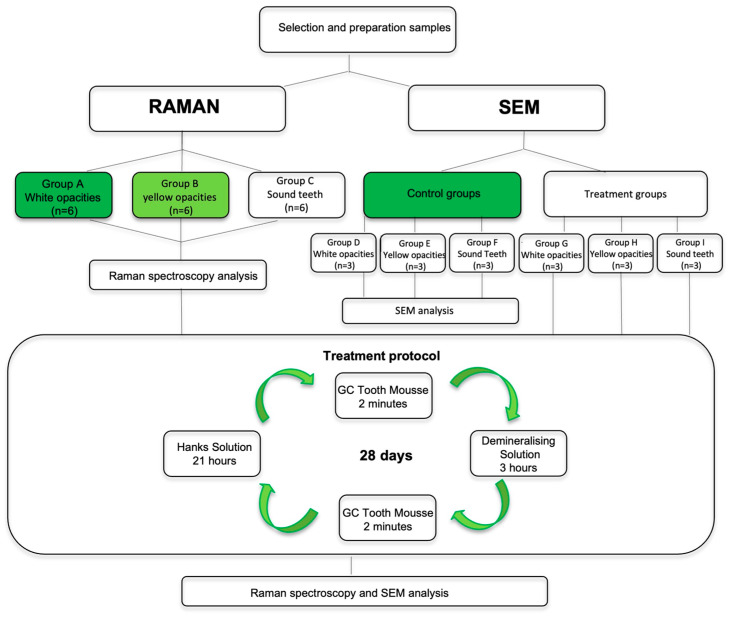
Scheme of the study design.

**Figure 2 biomedicines-10-03086-f002:**
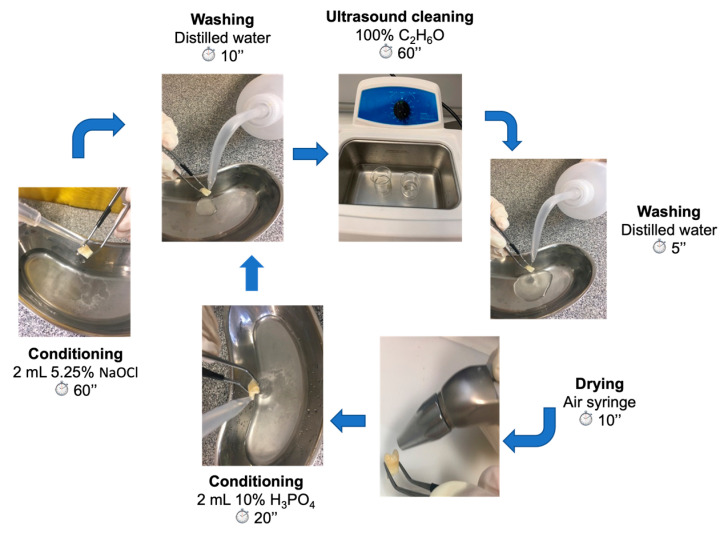
Protocol for debris removal and enamel prism revealing.

**Figure 3 biomedicines-10-03086-f003:**
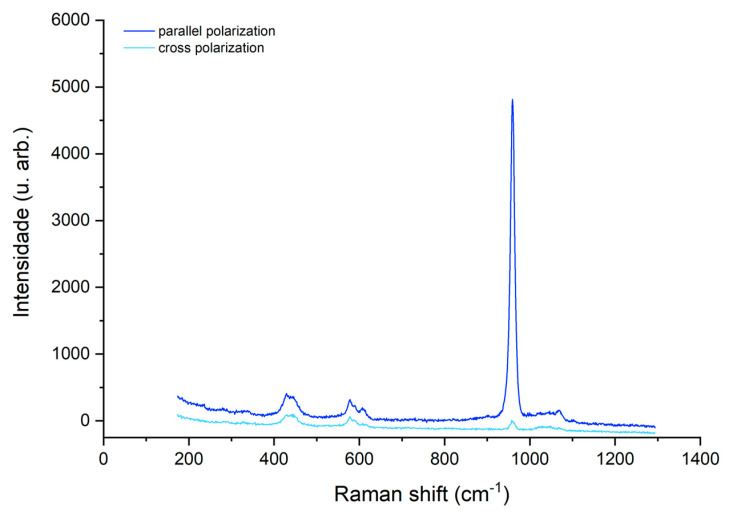
Spectra of a hypomineralized white opacity before treatment with GC Tooth Mousse.

**Figure 4 biomedicines-10-03086-f004:**
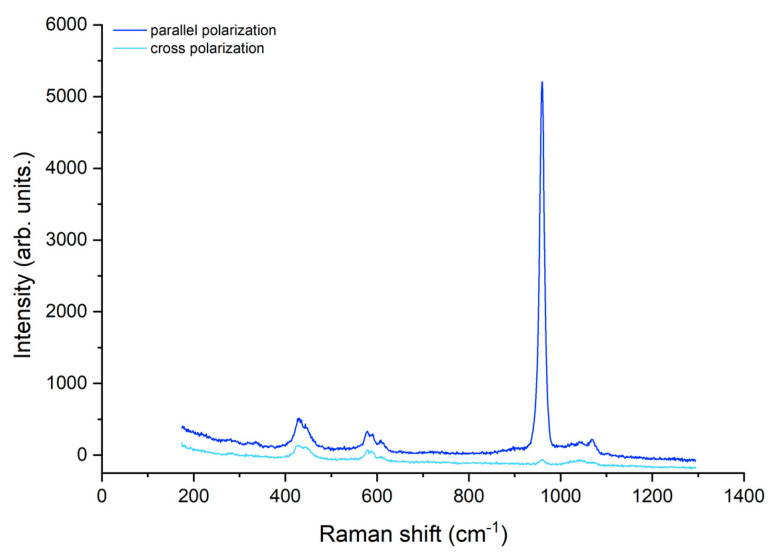
Spectra of a hypomineralized white opacity after treatment with GC Tooth Mousse.

**Figure 5 biomedicines-10-03086-f005:**
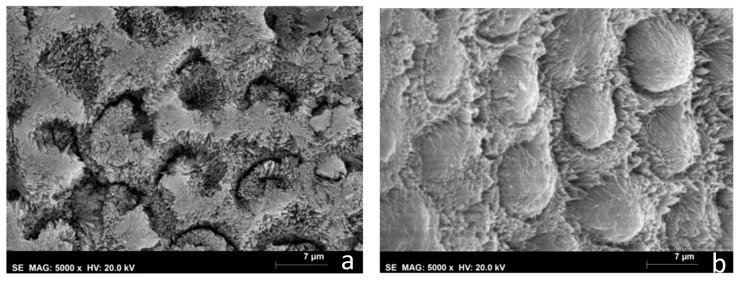
MIH specimens’ SEM photograph (5000×). (**a**) Group D—white hypomineralized opacities; (**b**) Group G—white hypomineralized opacities after CPP-ACP protocol.

**Figure 6 biomedicines-10-03086-f006:**
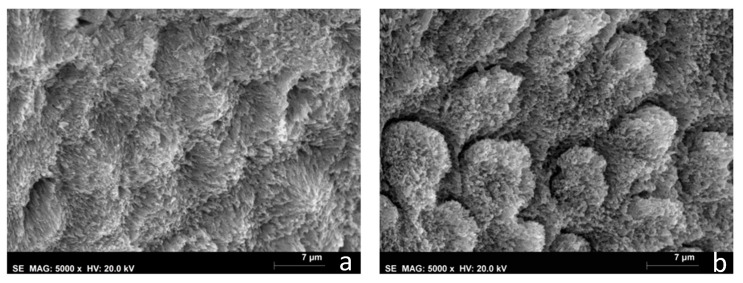
MIHs specimens’ SEM photographs (5000×). (**a**) Group E—yellow hypomineralized opacities; (**b**) Group H—yellow hypomineralized opacities after CPP-ACP protocol.

**Figure 7 biomedicines-10-03086-f007:**
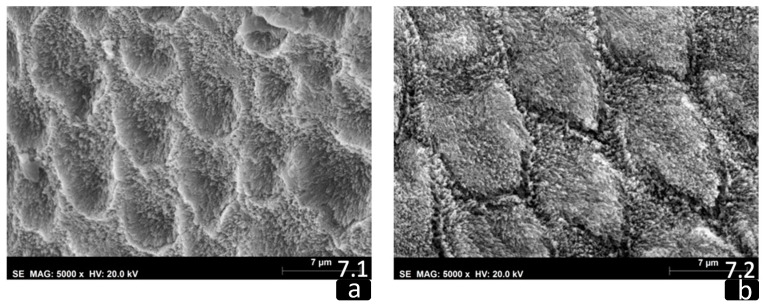
Sound specimens’ SEM photographs (5000×). (**a**) Group F—sound tooth; (**b**) Group I—sound tooth after CPP-ACP protocol.

**Table 1 biomedicines-10-03086-t001:** Mean depolarization ratio and standard deviations (SD) of the symmetric stretching band of phosphate values for the three groups.

Study Group	Depolarization Ratio Pre-Treatment (Mean± SD)	Depolarization Ratio Post-Treatment (Mean ± SD)	Significance Level (*p*)
A—White opacity teeth	0.029 ± 0.004	0.021 ± 0.003	0.004
B—Yellow opacity teeth	0.044 ± 0.004	0.037 ± 0.008	0.015
C—Sound teeth	0.019 ± 0.002	0.017 ± 0.003	0.5

## Data Availability

Not applicable.

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
