# Peer review of "Evaluation of the Efficacy of CPP-ACP Remineralizing Mousse in Molar-Incisor Hypomineralized Teeth Using Polarized Raman and Scanning Electron Microscopy—An In Vitro Study"

_biomedicines, 2022, doi:10.3390/biomedicines10123086_

Round 1

Reviewer 1 Report

Review of the manuscript entitled ‘Evaluation of the efficacy of CPP-ACP remineralizing mousse in Molar-Incisor hypomineralized teeth using polarized Raman and Scanning Electron Microscopy – an in vitro study’ by I. Cardoso-Martins et al. In this paper, the authors evaluate the efficacy of CPP-ACP tooth mousse in remineralizing MIH affected enamel using polarized Raman microscopy and scanning electron microscopy.

The research is interesting and the manuscript could be accepted after taking into account the minor comments. 

1.  I advise to use ‘Raman spectroscopy results’, ‘Raman spectroscopy analysis’, rather than ‘Raman results’, ‘Raman analysis’, also in Fig. 1.

2.     It is written ‘spectra were recorded without polarization’, however these results are not described.

Reviewer 2 Report

Dear Authors, 

Attached please find the PDF with the comments.

Good work

Round 2

Reviewer 2 Report

Dear Authors,

Thank you for providing the revised version of your work. I inform you that the modifications were correctly performed, therefore the manuscript is now suitable for publication.
Thank you for your submission to Biomedicines.